# Characterization of the Effect of a Novel Production Technique for ‘Not from Concentrate’ Pear and Apple Juices on the Composition of Phenolic Compounds

**DOI:** 10.3390/plants12193397

**Published:** 2023-09-26

**Authors:** José Carlos Teixeira, Catarina Ribeiro, Rodolfo Simôes, Maria João Alegria, Nuno Mateus, Victor de Freitas, Rosa Pérez-Gregorio, Susana Soares

**Affiliations:** 1LAQV-REQUIMTE, Chemistry and Biochemistry Department, Faculty of Sciences, University of Porto, Rua do Campo Alegre, 689, 4169-007 Porto, Portugal; josecarlos_1997@hotmail.com (J.C.T.); catarinaribeiro23@gmail.com (C.R.); rodolfodinisimoes@hotmail.com (R.S.); nbmateus@fc.up.pt (N.M.); vfreitas@fc.up.pt (V.d.F.); susana.soares@fc.up.pt (S.S.); 2SUMOL+COMPAL Marcas S.A, Estr. Portela 24, 2790-179 Carnaxide, Portugal; maria.alegria@sumolcompal.pt; 3Food and Agroecology Institute, University of Vigo, Campus As Lagoas, 32004 Ourense, Spain; 4Galicia Sur Health Research Institute (IISGS), Department of Chemistry and Biochemistry, Food and Health Omics Group, SERGAS-UVIGO

**Keywords:** phenolic compounds, fruit by-products, not-from-concentrate juice (NFC), centrifugal decanter, tangential filtration

## Abstract

The consumption of ‘not-from-concentrate’ (NFC) fruit juices can be a convenient and enjoyable way to incorporate the nutritional benefits and flavors of fruits into one’s diet. This study will focus on the effect of production of juices from apple and pear fruits, by using centrifugal decanter and tangential filtration, on the profile of polyphenols as a valuable source of bioactive compounds. Likewise, by-products from the juice industry were characterized in order to understand the high-value-added potential based on their composition of polyphenols. Briefly, apple and apple juice showed great contents of chlorogenic acid (0.990 ± 0.021 mg/g of DW), the dihydrochalcone phloridzin (1.041 ± 0.062 mg/g of DW), procyanidins (0.733 ± 0.121 mg/g of DW) and quercetin derivatives (1.501 ± 0.192 mg/g of DW). Likewise, the most abundant compounds in pear and pear juices were chlorogenic acid (0.917 ± 0.021 mg/g of DW), caffeoylquinic acid (0.180 ± 0.029 mg/g of DW), procyanidins (0.255 ± 0.016 mg/g of DW) and quercetin derivatives (0.181 ± 0.004 mg/g of DW). Both temperature and tangential speed affect the amount of phenolic compounds in fruit juices, highlighting the need to control the technological process to obtain a more nutritious/healthier beverage. Overall, NFC juices arise as a better option when compared with concentrated juices. Furthermore, the higher yield of phenolic compounds found in fruit pomace clearly open new ways for upcycling this fruit by-product as a high-value-added ingredient.

## 1. Introduction

Fruit is usually consumed fresh and also after processing for obtaining derived products. Among fruit-derived products, fruit juices are particularly significant, being one of the most consumed products worldwide. Fruit juices can be produced from a great variety of fruits and by different processing techniques. Recently, a greater focus has been placed on ‘not-from-concentrate juices’ (NFC juices), which are produced directly from fresh fruits without undergoing the concentration step involving the removal of water from the juice. Since NFC juices are made from fresh fruits, they may retain a higher level of nutrients compared to concentrated juices. They tend to contain more vitamins, minerals, and bioactive compounds that are naturally present in fruits. In this regard, NFC juices offer a wide range of flavors and nutritional profiles. In addition, regarding the nutritional composition, fruit juices have a significant amount of bioactive compounds such as phenolic compounds.

Phenolic compounds are widely known and have been vastly reported on due to their described bioactivities, such as chelating free radicals that can harm the lipids in cell membranes and/or cause nucleic acids to lose their structure [1,2,3,4,5]. Experimental evidence suggests that phenolic compounds not only inhibit the development of several diseases but also affect their spread, slow their advancement, and even aid in their recovery, probably due to their anti-inflammatory activities [6,7,8,9]. Structurally, phenolic compounds comprise a wide range of different compounds from simple phenol units to polyphenols, which can appear in nature glycosylated or acylated.

Considering the aforementioned, knowledge of the structural diversity of phenolic compounds in fruits and derived NFC juices is crucial for a global understanding of their general effect on human health. While the content of phenolic compounds in fruits depends on intrinsic and extrinsic factors, such as the cultivar, part of fruit, agronomic practices, environmental conditions, maturity stage, and harvesting, in NFC juices it also depends on the method of processing [10].

Among all the influencing factors, the technology applied to fruit plays a pivotal role in the phenolic compound’s composition of NFC juices. Fruit cell wall disintegration (cutting, grinding, etc.) leads to alterations in the phenolic compound’s profile due to enzymatic or chemical activity such as oxidation [9]. Indeed, studies regarding the direct effect of the processing technology on the profile of phenolic compounds and their impact on human health is a priority among food industry and food science.

This study, thus, aims to understand the effect of fruit processing on the profile of phenolic compounds in NFC pear- and apple-derived juices. Firstly, the effect of centrifugation, filtration and temperature to produce NFC will be studied. In parallel, the profile of phenolic compounds of the by-products resulting from fruits’ processing will be also characterized, under a circular economy and sustainability umbrella.

## 2. Results

### 2.1. Total Polyphenol Content by Folin–Ciocalteu

#### 2.1.1. Apple Matrices and By-Products

The total phenolic content analyzed using Folin–Ciocalteu (FC-TPC) is displayed in Figure 1. As shown, the total phenolic content (FC-TPC) was quantified in apple (A), apple puree (with (PWU) and without (P) homogenization in the colloidal milling in Urschel^®^), apple pomace (with (POWU) and without (PO) colloidal milling), clarified juices at different temperatures (40 °C and 60 °C) and turbid juices obtained after centrifugal decantation at two different centrifugal forces/speeds (35 or 10 rpm) following the different conditions displayed in Section 4. As shown, fresh fruit had the highest FC-TPC (29.240 ± 0.7361 mg/g of DW) which decreased after crushing treatment. Crushing the apple fruit to obtain purees led to an overall decrease of approximately 7% of FC-TPC. The colloidal milling did not significantly affect the FC-TPC of the apple puree (P 27.157 ± 0.157 and PWU 27.044 ± 0.635 mg/g of DW). In parallel, while the pomace accounted for approximately 25% less FC-TPC when compared with fruit, it arose as a rich source of phenolic compounds.

Results, thus, indicate that processing methods used to obtain ‘not from concentrated’ fruit juices significantly reduces the FC-TPC. This reduction depends on the treatment. Namely, Figure 1 shows that both centrifugation forces and clarification temperature significantly affected the FC-TPC in turbid or clarified apple juices, respectively. In this regard, 35 rpm for centrifugal decantation and 40 °C for tangential filtration led to a significantly higher FC-TPC, with values ranging from 0.505 ± 0.019 mg/g of DW in TJF35 to 0.264 ± 0.007 mg/g of DW in TJ10 and from 0.5414 ± 0.015 mg/g DW for CJ40 to 0.381 ± 0.010 mg/g DW.

#### 2.1.2. Pear Matrices and By-Products

Figure 2 shows the FC-TPC quantified in pear (PE), pear puree (with (PPWU) and without (PP) homogenization in the colloidal milling Urschel^®^, Urschel laboratories, Chesterton, Indiana), pear pomace (with (POPWU) and without (POP) colloidal milling), clarified juices at different temperatures (40 °C and 60 °C) and turbid juices obtained after centrifugal decantation with two forces (35 or 10 rpm). As shown, the amount of FC-TPC quantified in pear was significantly lower when compared with puree or pomace despite the processing technology used. Indeed, the industrial crushing of pear (PP) resulted in roughly double the concentration of FC-TPC in comparison with the unprocessed fruit. However, for puree, the colloidal milling in Urschel^®^ resulted in a decrease in the FC-TPC. Likewise, the pomace (originating as a by-product after passing the puree through a centrifugation decanter to obtain the turbid juice) also had a slightly significantly higher FC-TPC concentration when compared with pear.

As shown in Figure 2, the differential speed clearly influenced the FC-TPC quantified in turbid pear juice. Indeed, a higher speed (35 rpm) lead to a higher recovery of phenolic compounds; while using 10 rpm we did not find any significant difference compared with clarified juices. Likewise, the filtration temperatures did not significantly affect the FC-TPC in clarified pear juices.

### 2.2. Phenolic-Compound Characterization and Identification

#### 2.2.1. Apple Matrices and By-Products

The profile of the phenolic compounds in apple, juice and byproducts was determined using HPLC-UVvis and the identification of the phenolic compounds present was performed using LC-MS/MS (Table 1).

Table 1 shows the identification of major phenolic compounds in apple, apple-derived products and their byproducts. Thirty three phenolic compounds were properly identified while some minor compounds remain unidentified.

Looking at the chromatogram and the identification data, the major phenolic compounds are chlorogenic acid (peak 11), followed by the flavan-3-ols, namely procyanidin C1 trimer (peak 14), epicatechin (peak 21), procyanidin trimer (peak 22), the phenolic acid coumaroylquinic acid (peak 23) and the hydrochalcone phlorizdin (peak 47). Moreover, apple contained a minor amount of flavonols, mainly quercetin derivatives.

In addition, phenolic compounds were quantified using a chlorogenic acid calibration curve. The sum of each compound was obtained as a measure of the total phenolic compounds quantified using liquid chromatography, HPLC-TPC. As displayed in Figure 3, quantification using HPLC analysis showed that crushing (P) and colloidal milling (PWU) did not affect the HPLC-TPC quantified. While pomace samples, the main by-products of fruit juice industry, showed a significant reduction in their HPLC-TPC level in comparison to apple and puree, it still had a considerable HPLC-TPC, representing almost 40% of the fruit’s polyphenol composition. However, in pomace, the colloidal milling with Urschell led to a lower HPLC-TPC.

#### 2.2.2. Pear Matrices and By-Products

The profiles of the phenolic compounds in pear, its derived products and its byproducts were determined using HPLC-UVvis and the identification of the phenolic compounds present was performed using LC-MS/MS (Table 2). Forty phenolic compounds were successfully identified.

Looking at the characterization and identification data, chlorogenic acid is the major polyphenol found in pear (peak 13). Many other phenolic compounds were found to be present in pear but in very low concentrations in comparison to chlorogenic acid.

HPLC-TPC was quantified in the same manner to apple and its by-products, and the results are displayed in Figure 4. Quantification using HPLC-TPC analysis showed a higher recovery of phenolic compounds in the puree when compared with the whole pear. Colloidal milling did not affect the phenolic content in the puree but colloidal milling using an Urschell resulted in a lower amount of phenolic compounds in the pomace obtained. Likewise, the tangential filtration speed clearly affected the amount of phenolic compounds present in turbid juices, with a higher content obtained for 35 rpm. The clarification temperature did not significantly affect the amount of phenolic compounds in pear juices.

## 3. Discussion

This work was devoted to understand how different industrial processing technologies (colloidal milling, different forces of centrifugal decantation and different temperatures of tangential filtration) affect the phenolic content of ‘not from concentrated’ fruit juices derived from apple and pear, looking at all products in the stages between the raw fruit and the final production of the different types of juice. To evaluate this, phenolic compounds were characterized by means of spectrophotometric, chromatographic, and mass spectrometry methods and the differences depending on the sample and processing methods were highlighted.

In this study, 47 different phenolic compounds were detected in apple, with 32 properly identified in apple fruit and its subproducts. To date, more than 60 phenolic compounds have been identified in apple, highlighting the flavonoid’s quercetin and the dihydrochalcone phloridzin [6,9]. Qualitative differences have been found in the profile of phenolic compounds characterized in apple. While some authors [11] identified 20 different compounds, other authors have also identified 11 [12] and 7 individual phenolics [13]. This high variability can be attributed to genetic factors (apple variety) or agronomic and environmental conditions.

Besides the qualitative differences found in the profile of phenolic compounds in apple, the main objective of this work was to elucidate the effect of industrial processing to obtain “not from concentrated” juices.

In apple matrices, the FC-TPC was higher in apple fruit and decreased upon each processing method. This has been previously observed, and the extent of this decrease depends on the processing method [14]. The phenolic compounds identified in these apple samples are oxidation-sensitive and have been identified as polyphenoloxidase substrates. This means that oxidation starts immediately after the apples are crushed into a puree. However, the puree production with or without milling did not affect the FC-TPC. While the first step to producing juices (puree) did not significantly affect the overall phenolic composition, the turbid and clarified juices showed a decrease of 1 to 4% of its initial phenolic content in the soluble fraction, a significantly lower decrease when compared with similar studies. In previous studies [14], the levels of flavonoids and chlorogenic acid in the juice reduced to between 50% (chlorogenic acid) and 3% (catechins). Indeed, the final HPLC-TPC of these juices was approximately 10-fold higher than reported previously for other commercial apple juices [15]. Furthermore, it was not possible to uncover the phenolic content in the insoluble fraction of the turbid juices as well as the non-extractable phenolic compounds. While hydrolysis was performed in aggressive conditions, with acid and basic conditions at high temperatures, these conditions can produce very small fragments, as observed using LC-MS analysis (Appendix A), making its identification impossible [16].

Overall, the total phenolic compounds significantly decreased during apple processing to obtain juice. Nevertheless, the processing conditions were verified to have affected the total amount of phenolic compounds in the final product. Namely, the higher the differential filtration speed was (35 rpm) in the turbid juice (TJ35), the higher the phenolic content was. For differential filtration, juices are fed into a centrifugal decanter, which causes the solids to settle to the bottom while the liquid is removed from the top when the juice is spun at high speeds. The value of the differential speed varies inversely with the residence time of the solids inside the decanter: the lower the differential speed, the greater the residence time of the solids inside the decanter. Results show that a higher differential speed led to an increase in the phenolic content, perhaps due to solids remaining in the decanter for less time, thus being unable to pull as many phenolic compounds from the liquid/soluble fraction. In addition, the tangential filtration process (that consists of a membrane filtration system that removes any remaining solids or impurities from the juice) at a lower temperature seems to affect, to a small extent, the phenolic content, since the clarified juice derived from the turbid juice TJ35 (CJ40) only had a small decrease in its phenolic content. However, when this process occurs at a higher temperature (60 °C, CJ60) it causes a large reduction in the phenolic content.

The pomace byproduct, obtained with or without colloidal milling, still retained a significant amount of phenolic compounds, as has already been reported. This is easily explained by the significant amount of phenolic compounds that are covalently bound to cell walls. This value-added byproduct has been already applied for different purposes in the food industry and pharmaceutical industry [17,18].

It should also be noted that the overall amount of phenolic compounds generally decreased while specific qualitative differences were not observed, meaning that it maintained its chromatographic profile. Among the identified phenolics, chlorogenic acid was found to be the most abundant phenolic compound in all the samples. Apple showed great contents of chlorogenic acid (0.990 ± 0.021 mg/g of DW), procyanidins (0.733 ± 0.121 mg/g of DW) and quercetins (1.501 ± 0.192 mg/g of DW). In apple and its byproducts, phloridzin (1.041 ± 0.062 mg/g of DW in apple) was also identified.

Likewise, pear had 72 phenolic compounds identified, with 54 compounds being characterized for the first time in pears [19]. In this study, 55 different compounds were detected and 40 were properly identified. Among them, not only phenolic compounds but also organic acids were properly identified after phenolic-compound extraction.

Contrary to the apple matrices, TPC-FC was lower in whole pears and increased upon the crushing process to obtain the puree (PP). One possible explanation for this observation is related to the ripening stage of pears. Overripening has already been shown to significantly decrease the extraction of phenolics due to the binding of phenolics to the cell walls [20]. Upon pear crushing and milling, cell walls and some of its interactions could be ruptured; therefore this could facilitate the extraction of phenolic compounds. Then, HPLC-TPC decreased after processing. As observed for the apple matrix, processing methods have been shown to decrease the phenolic content in pear juices which could be once again possibly explained by the oxidation sensitivity of phenolic compounds. Both turbid and clarified juices have as low as 2% of the puree phenolic content of that in the soluble fraction. Nevertheless, as in the case of apple juice, the final HPLC-TPC of pear juices was nearly 10-fold higher than that previously reported by other commercial pear juices [15]. As mentioned previously, it was not possible to uncover the phenolic content in the insoluble fraction of the turbid juices. In the case of pears, and considering its cell structure and associated difficulties of the extraction of phenolic compounds, this issue has some relevance. Further studies are required to properly understand the impact on non-extractable phenolic compounds. Conversely to the results found in apple, the differential filtration and tangential filtration processes result in turbid and clarified juices with the same phenolic content, regardless of the conditions used (speed and temperature).

Similar to apple pomace, pear pomace still retains a significant amount of phenolic compounds, which has already been reported [21]. This is easily justified by the significant amount of phenolic compounds that are covalently bound to cell walls.

The phenolic profile of pear matrices was also analyzed. Once again, the identified phenolic compounds represented a small percentage of the total compounds, as evidenced by comparing the quantifications obtained using the Folin–Ciocalteau method and using liquid chromatography. These differences were much higher for the puree and pomace matrices than for the pear fruit and juices. The fact that Folin–Ciocalteau and HPLC-UVvis yielded comparable quantifications of total phenolics in pear fruit may be explained by the fact that the quantified phenolics are predominantly unbound and possess a low molecular weight, such as phenolic acids, as opposed to phenolic compounds that are bound to the cell walls, which are more complex and have a higher molecular weight [21]. Puree and pomace matrices possibly contain more complex phenolic compounds with high-molecular-weight phenolics that were released upon crushing and milling, as explained before and as evidenced by the increase in the HPLC-TPC in the puree. These complex and high-molecular-weight phenolics are difficult analyze and identify using LC-MS/MS.

Once again, chlorogenic acid was the most abundant phenolic compound in all samples. In pear, the most abundant compounds were chlorogenic acid (0.917 ± 0.021 mg/g of DW), caffeoylquinic acid (0.180 ± 0.029 mg/g of DW), procyanidins (0.255 ± 0.016 mg/g of DW) and quercetin derivatives (0.181 ± 0.004 mg/g of DW).

In summary, comparing apple and pear products, apple products were found to have more phenolics. In addition, some processing methods seemed to produce the same effect in both fruit matrices, namely, differential filtration and tangential filtration, with a high differential filtration (35 rpm) and low tangential filtration temperature (40 °C) resulting in the highest phenolic content. However, it should be highlighted that these results refer to the soluble and extractable phenolic compounds. In fact, turbid juices have been shown to have more phenolics than clarified juices [22], but this higher content is probably due to bounded compounds. So, the differential filtration and tangential filtration conditions used in this study seem to result in more eventually (bio)accessible phenolic compounds. As discussed previously, the NFC juices produced by the referred approaches had a high content of phenolic compounds when compared to the literature, and above all, could be better than juices produced by concentration, because they preserve their natural appearance and fresh flavor [23]. Regarding the milling process, its effect seems to depend on the fruit and food matrix. In puree, it gives a different phenolic content in the pear but not in the apple matrix; in pomace, it does not affect the phenolic content regardless of the fruit matrix. In each matrix, the major compound is chlorogenic acid.

Overall, apple- and pear-derived NFC juices and byproducts are a good source of phenolic compounds. The findings of the current study unequivocally demonstrate that processing can significantly affect a product’s potential health advantages. Therefore, researching how different processing techniques affect the quantities of bioactive substances, as well as the biological activities of various fractions produced during processing, will give researchers tools for determining the best ways to enhance a product’s healthfulness.

The integration of processes, including its technical, economic, social, and environmental aspects, is anticipated to be further explored shortly due to concerns about transitioning from linear to circular economies. This will likely lead to more realistic, greener platforms for using food byproducts, such as apple and pear pomaces, to their full potential [6]. Apple pomace is a low-cost source of phytochemicals and bioactive compounds, such as polyphenols, dietary fiber, pectin, triterpenoids, and volatiles. Pear pomace has a high nutritional value, with reasonable amounts of sugars, amino acids, and dietary fiber, and contains other nutritional and bioactive components such as polyphenols. The pomace of apples and pears in the study showed more than 3 and 2 mg/g of DW, respectively. Its byproducts can be used for many applications like as sources of natural polymers for drug delivery and sources of phytochemicals and nutraceuticals, such as those used in cosmetic manufacturing. Pomaces are also high in dietary fiber, and their intake helps prevent and control chronic diseases and improve digestive health.

As it is known, in apple, non-extractable compounds correspond to 6,7% of the total phenolic content [24], and these values are not very different for pear [25]. Fruits and vegetables may contain phenolic compounds in the form of non-extractable conjugates, soluble conjugates (such as glycosides and fatty acid esters), and extractable conjugates. The latter can either be chemically or physically attached to the proteins or polysaccharides that make up the cell wall via ester, ether, or glycosidic linkages in food matrices and intact cells. However, if these substances are still conjugated to glycosides, fatty acid esters, or other types of bonds, they will not be able to be analyzed using HPLC-UVvis, which may be one explanation for the discrepancy between the HPLC-UVvis and Folin observed. On the other hand, when alcohol or hydro-alcohol extractions are utilized, procyanidins are typically overestimated; the majority of it are not removed and remain in the insoluble area of the cortex. Additionally, because polymeric forms do not provide well-resolved peaks on chromatograms, their estimate using HPLC-UVvis following extraction is still insufficient [20].

## 4. Materials and Methods

### 4.1. Chemicals

All reagents used were characterized by analytical grade. Reagents for the extraction of phenolic compounds and HPLC-DAD analysis were multisolvent and LC-MS quality was chosen for identification purposes using mass spectrometry. Acetonitrile (99.8%) was purchased from Panreac Quimica (Barcelona, Spain); formic acid (99%) and methanol (99.8%) were purchased from ChemLab (Zedelgem, Belgium); diethyl eter (99%) and ethyl acetate (99.9%) were purchased from Fisher chemical (Geel, Belgium); Folin reagent and sodium hydroxide were purchased from LabChem (ACP Chemicals Inc., Zelienople, PA, USA); chloridic acid was purchased from Honeywell Fluka (Buchs, Switzerland); sodium carbonate was purchased from Sigma-Aldrich (Sigma Aldrich, St. Louis, MO, USA); and water was obtained from a Milli-Q^®^ Direct Water Purification System. Reference standards including those of gallic acid, protocatechuic acid, chlorogenic acid, caffeic acid, catechin, epicatechin, epicatechin gallate, quercetin, quercetin-3-O-glucoside, phlorizin, phloretin, p-coumaric acid, ferulic acid, 4-Caffeyolquinic acid, and kaempferol were acquired from Sigma-Aldrich (Sigma Aldrich, St. Louis, MO, USA).

### 4.2. Industrial Processing

All fruit, purees, juices and byproducts samples were provided by SUMOL+COMPAL, a Portuguese company. Apple and pear samples were obtained and were crushed with or without Urshell, producing purees. After this, the purees and the corresponding supernatant were centrifugated to juices, and the pellets to pomaces. The method of centrifugation and speed was changed to obtain juices with different turbidities and also different pomaces. After decantering, tangential filtration was applied to turbid juices with different temperatures to give different clarified juices, as shown in Figure 5.

### 4.3. Samples

The purees and pomaces were dried and stored at −20 °C until analysis. Juices were stored also at −20 °C. Table 3 shows all samples and their processing methods, and also the abbreviations used within this document.

### 4.4. Phenolic-Compound Extraction from Fruit Juices

After defrosting at 4 °C, 5 mL of each juice was taken. Turbid juices were first centrifugated on a Hettich Universal 320 R refrigerated Benchtop Centrifuge, and pellets were stored at −20 °C until extraction. The supernatants of turbid juices and 5 mL of each clarified juice were used to perform SPE (solid phase extraction), with C18 solid phase (35 cc Vac cartridge—Waters, MA, USA) for removing sugars with water, and the phenolic compounds were recovered using elution with 1 mL of methanol, 1 mL of methanol/acetone (1:1) and finally 1 mL of methanol/acetone (1:1) acidulated with HCl. Hereafter, the organic solvent was evaporated in a rotavapor R-300 Buchi under vacuum at 37 °C until 1 mL and 1 mL of miliQ water was added. All extracts were stored at −20 °C until analysis and were analyzed within two days maximum. Figure 6 summarizes de extraction procedure.

### 4.5. Phenolic-Compound Extraction from Fruits

Free phenolics were extracted from fruit samples using conventional maceration by using hydroalcoholic solutions [26]. For this purpose, 30 mL of cold methanol/water (80:20) with 1% of formic acid were added to 0.5 g of freeze-dried fruit samples and were processed with T25-Ultra-turrax (IKA-Labortechnik^®^, Staufen, Germany) for 3 min at 16,000 min^−1^. Then, the mixture was centrifuged 20,000× *g*, 10 min at 4 °C on a Hettich Universal 320 R refrigerated Benchtop Centrifuge. The extraction process was repeated twice.

The supernatants of each extraction were combined and evaporated to 1 mL in a rotavapor R-300 Buchiunder vacuum at 37 °C and dissolved in 1 mL of miliQ water. All extracts were stored at −20 °C until analysis and were analyzed within two days maximum.

### 4.6. Extraction of Bound Phenolic Compounds 

The pellet resulting from the centrifugation of turbid juices and pomaces were subjected to a sequential base and acid hydrolysis [26]. For base hydrolysis, the remaining residues were hydrolyzed with 15 mL of NaOH 2M, for 4 h at room temperature (around 22 °C) under a stream of argon. The resulting slurry was acidified to pH2.0 with HCl 6 M and then centrifuged at 20,000× *g* for 10 min on a Hettich Universal 320 R refrigerated Benchtop Centrifuge. The supernatant was extracted with diethyl ether/ethyl acetate (1:1, *v*/*v*) three times and organic fractions were stored at −20 °C. For subsequent acid hydrolysis, the remaining pellet of basic hydrolysis was incubated with 15 mL of HCl (2M), heated at 85 °C for 1 h, brought to pH 2 with NaOH (6M), centrifuged at 20,000× *g* for 10 min on a Hettich Universal 320 R refrigerated Benchtop Centrifuge, and then the supernatant was extracted with diethyl ether/ethyl acetate (1:1, *v*/*v*) three times. Organic fractions, from acid and basic hydrolysis were combined and evaporated to 1 mL, which was recovered and dissolved on 1 mL of miliQ water. These fractions were composed of the bound phenolic compounds, since there was no possible way to identify the bounded phenolics. The table containing the MS/MS results are included as Appendix A.

### 4.7. Determination of Total Phenolic Content of the Extracts

The total phenolic content of the fresh fruit, puree, pomace and fruit juices were estimated using Folin–Ciocalteu. The reaction mixture was prepared by mixing 3.75 µL of each extract (fruit, juices, purees and pomaces) with 18.75 µL of Folin–Ciocalteu phenol reagent, and 125 µL of water in 96 plates. The mixture was shaken for 30 s, and 75 µL of Na_2_CO_3_ and 152.5 µL of water was added. The mixtures were shaken again and left at room temperature for 30 min in the dark. The mixtures were transferred to a 96-well plate and analyzed using absorbance at 750 nm in a plate reader (Biotek, Berthold Technologies, Bad Wildbad, Germany). For quantification purposes, a calibration curve with chlorogenic acid was performed using the same procedure. Extract concentrations were obtained by means of mg chlorogenic acid equivalents (CAE) per mL of fresh weight (FW) (mg CAE/mL FW) for juice samples, and mg CAE per g of dry weight (DW) (mg CAE/g DW) for fruits, puree, and pomaces, in a range from 1 to 50 mg/g.

### 4.8. Characterization of Phenolic Compounds Using HPLC-UVvis and Identification Using LC-MS/MS

Phenolic-compound quantitative analysis was performed using HPLC-DAD (Vanquish Thermo Fischer Scientific, Waltham, MA, USA), equipped with an Agilent Poroshell 120, C18 reverse-phase column (250 × 4.6 mm, 2.7 μm particle diameter). For apple (juices, puree’s and pomace’s), the solvents were A) 0.5% formic acid in water and B) 0.5% formic acid in acetonitrile, with a gradient of 5% of eluent B at 0 min, 20% B at 22 min, 25% B at 52 min, 100% B at 57, and 100% B at 62 min, at a flow rate of 0.5 mL.min^−1^ and detection levels of 280 nm (flavan-3ols, phenolic acids and dihydrochalcones) and 360 nm (flavonols). For pear (juices, purees and pomaces), the solvents were A) 1% formic acid in water and B) 100% acetonitrile, with a gradient of 3% of eluent B at 0 min, 50% B at 60 min, 100% B at 65 min, and 100% B at 75 min, and a flow rate of 0.5 mL.min^−1^ and detection levels of 280 nm and 360 nm. Reference standards including gallic acid, protocatechuic acid, chlorogenic acid, caffeic acid, catechin, epicatechin, epicatechin gallate, quercetin, quercetin-3-O-glucoside, phlorizin, phloretin, p-coumaric acid, ferulic acid, 4-Caffeyolquinic acid, and kaempferol were used to compare the retention time to help identification. The identified compounds were quantified in equivalents of quercetin, epicatechin, chlorogenic acid or phlorizdin depending on the families (favonols, flavan-3ols, phenolic acids or dihydrochalcones, respectively). Therefore, quantifications were expressed by mg of CAE upon peak integration using Chromeleon™ 7.2. Chromatography Data System (CDS) Software (Waltham, MA, USA). The calibration curve were used in the range of 0.01 to 0.06 ppm for epicatechin, quercetin and phlorizdin and up to 0.25 ppm for chlorogenic acid.

The identification of extractable phenolic compounds was performed using LC-MS/MS, using the same HPLC conditions as previously described.

Analysis was performed on an Orbitrap TM Exploris 120 (Thermo Fischer Scientific, Bremen, Germany) controlled using Orbitrap Exploris 120 Tune Application 2.0.182.35 and Xcalibur 4.4.16.14. The capillary voltage of the electrospray ionization source (ESI) was set to 3.5 kV. The capillary temperature was 300 °C. The sheath gas and auxiliary gas flow rate were 50 and 10 (arbitrary units as provided by the software settings). The resolution of the SIM MS scan was 30,000. Collision energy settings of 30 V were selected, and MS/MS was performed on HCD.

Spectra were recorded in both negative- and positive-ion mode between *m*/*z* 100 and 2000. The mass spectrometer was programmed to perform a series of three scans: a full mass, a zoom scan of the most intense ion in the first scan, and an MS-MS of the most intense ion using relative collision energies of 30 V. The analysis was performed with Xcalibur version 2.2 software (Thermo Scientific, Waltham, MA, USA). The identification of phenolic compounds was performed by using the already referredd standards and by comparing the obtained fragmentation pattern and exact mass with previous data available in the literature, and databases like phenol explorer and compound discovery.

### 4.9. Statistical Analysis

All assays were performed in three independent triplicates and on different days as well. The mean values and standard deviation were evaluated using one-way analysis (ANOVA). All statistical data were processed using GraphPad Prism version 9.0 for Windows (GraphPad Sofware, San Diego, CA, USA).

## 5. Conclusions

Polyphenols are important compounds found in some fruits, such as apples and pears, which offer numerous health benefits. Processing methods like tangential filtration and centrifugal decanter, to produce NFC juices, can affect the amount of polyphenols present in the juice. While each method has its advantages and disadvantages, NFC juices appear to be the best option for preserving the natural flavor and nutritional value of the fruit, including the polyphenols.

Despite the knowledge that turbid juices have more polyphenols than clarified juices, results obtained in this paper have highlighted that tangential filtration using low temperatures to clarify, like 40 °C, can obtain a juice with a similar polyphenol content as turbid juices produced using centrifugal decanter at a high differential speed. However, further studies are needed to understand how far the increase in differential speed allows the growth of polyphenols, because even without statistical difference, TJP35 has a worse polyphenol content than TJP20. So high differential velocities can lead to the loss of polyphenols, therefore it is important to understand the best speed of operating.

In conclusion, NFC juices produced using centrifugal decanter and tangential filtration are proper options to obtain juices with a higher amount of polyphenols.

Under this context, the results presented herein contribute to a better understanding of the impact on the processing methods used to obtain the apple or pear juices on its content of phenolic compounds, helping in the design of novel processing technologies for obtaining fruit juices with the maximum amount of phenolic compounds.

## Figures and Tables

**Figure 1 plants-12-03397-f001:**
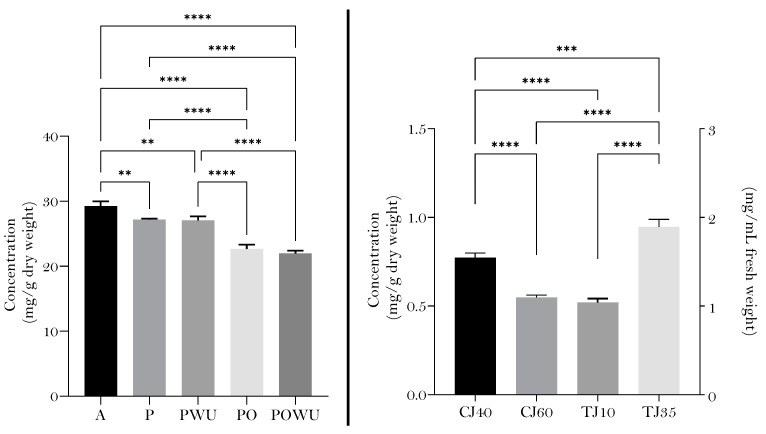
Total phenolic content of apple, apple-derived products and byproducts determined using Folin–Ciocalteu in equivalents of chlorogenic acid (left graphic: A—apple, P—puree without colloidal milling, PWU—puree with colloidal milling, PO—pomace without colloidal milling, POWU—pomace with colloidal milling; right graphic: TJ10RPM—turbid juice using a differential speed of 10 RPM, TJ35—turbid juice using a differential speed of 35 RPM, CJ40—clarified juice at 40 °C, CJ60—clarified juice at 60 °C); ** *p*-value of 0.01, *** *p*-value of 0.001, **** *p*-value of 0.0001.

**Figure 2 plants-12-03397-f002:**
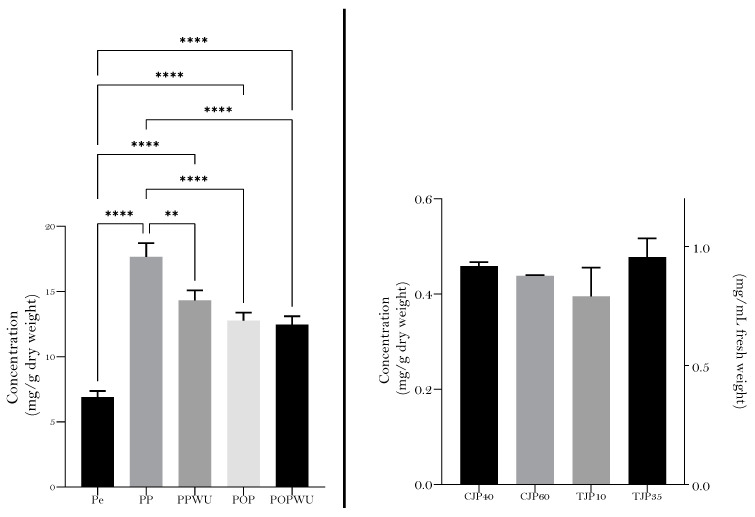
Total phenolic content of pear, pear-derived products and byproducts determined using Folin–Ciocalteu in equivalents of chlorogenic acid (left graphic: PE—pear, PP-pear puree, PPWU—pear puree with colloidal milling using Urschel^®^, POP—pear pomace, POPWU—pear pomace with colloidal milling using Urschel^®^, TJP10—turbid pear juice obtained at 10 rpm of differential speed during centrifugal decantation, TJP35—turbid pear juice obtained at 35 rpm of differential speed during centrifugal decantation, CJP40—clarified pear juice at 40 °C tangential filtration, CJP60—clarified pear juice at 60 °C tangential filtration; ** *p*-value of 0.01, **** *p*-value of 0.0001.

**Figure 3 plants-12-03397-f003:**
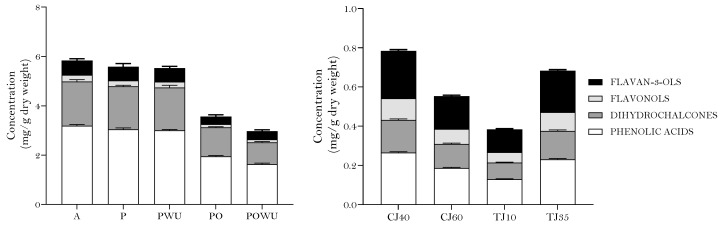
Phenolic content of apple, apple-derived products and byproducts determined using HPLC in equivalents of chlorogenic acid (left graphic: A—apple, P—puree without colloidal milling, PWU—puree with colloidal milling, PO—pomace without colloidal milling, POWU—pomace with colloidal milling; right graphic: TJ10RPM—turbid juice using a differential speed of 10 RPM, TJ35—turbid juice using a differential speed of 35 RPM, CJ40—clarified juice at 40 °C, CJ60—clarified juice at 60 °C).

**Figure 4 plants-12-03397-f004:**
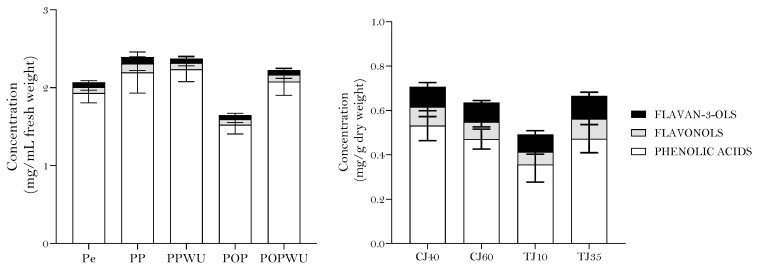
Phenolic content of pear, pear-derived products and byproducts determined using HPLC in equivalents of chlorogenic acid (left graphic: PE—pear, PP—puree without colloidal milling, PPWU—puree with colloidal milling, POP—pomace without colloidal milling, POPWU—pomace with colloidal milling; right graphic: TJ10RPM—turbid juice using a differential speed of 10 RPM, TJ35—turbid juice by the differential speed of 35 RPM, CJ40—clarified juice at 40 °C, CJ60—clarified juice at 60 °C).

**Figure 5 plants-12-03397-f005:**
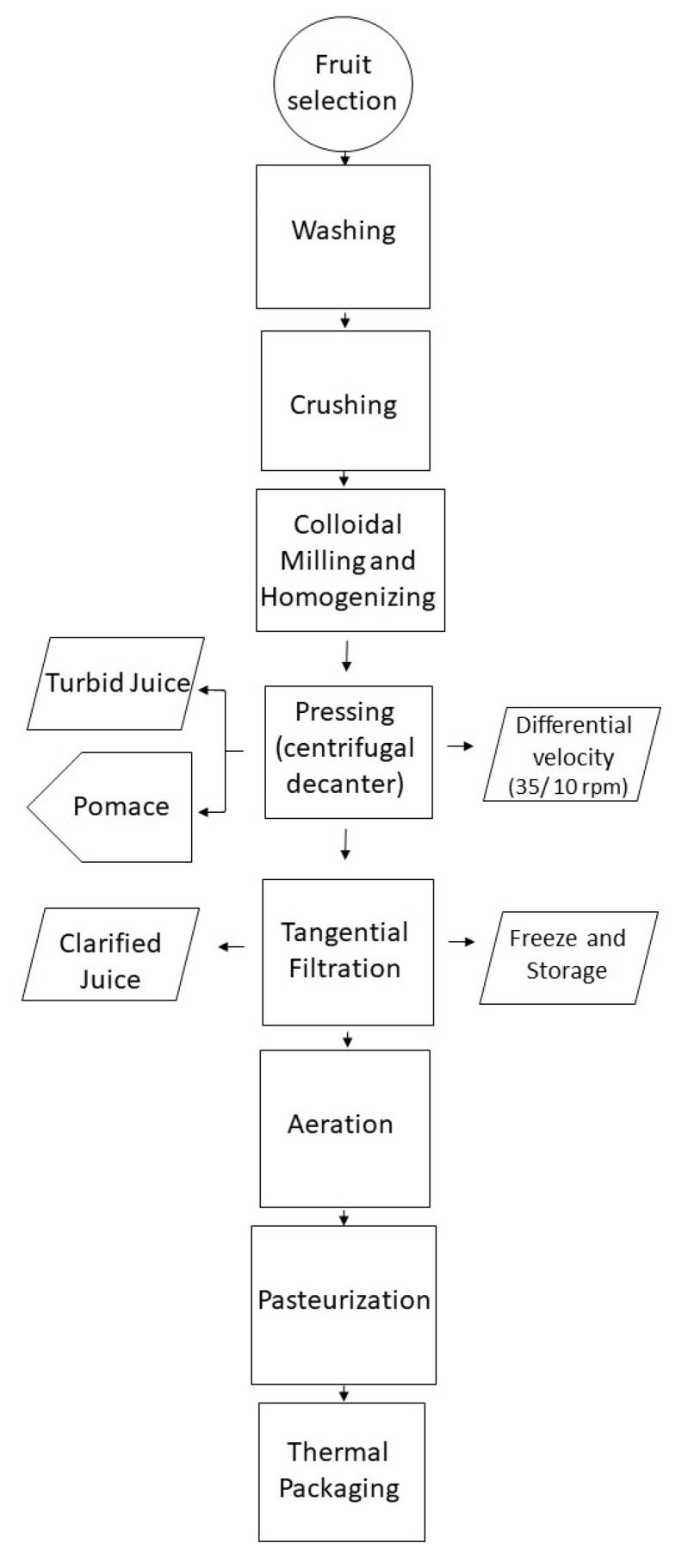
Industrial processing of apple and pear fruits.

**Figure 6 plants-12-03397-f006:**
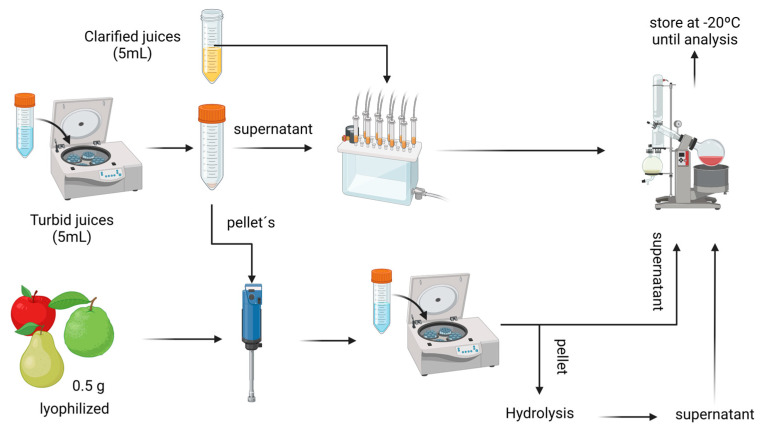
Processing used to obtain phenolic compound extracts from fruit samples, and turbid and clarified juices. Ultraturrax was used to assist the extraction process of both freeze-dried fruits and the resulting pellet from turbid juices.

**Table 1 plants-12-03397-t001:** Identification of phenolic compounds in apple matrices by using LC-ESI-MS/MS.

Proposed Compounds	RT (min)	Ionization (ESI^+^/ESI^−^)	Molecular Ion	MS/MS
Procyanidin	7.57	[M–H]^+^	867.21	127, 135, 163, 123
Procyanidin B3 Dimer	10.80	[M–H]^−^	577.13	289, 407, 125
Caffeic acid derivative	11.28	[M–H]^+^	163.00	89, 117, 135
Coumaric acid	11.73	[M–H]^+^	420.16	165, 123, 147
Procyanidin B1 trimer	12.21	[M–H]^−^	865.20	125, 289, 407, 287
Chlorogenic acid	12.39	[M–H]^−^	353.09	191
Caffeoylquinic acid	12.86	[M–H]^+^	355.10	163, 135, 145, 117, 89
Catechin	13.29	[M–H]^−^	289.07	245, 205, 109, 125
Procyanidin C1 Trimer	13.51	[M–H]^−^	865.20	125, 289, 407, 161
Procyanidin B2 Dimer	13.68	[M–H]^−^	577.13	289, 407, 125
Tetramer of procyanidin	13.90	[M–H]^+^	1155.00	135, 127, 123, 245, 163
Caffeic acid	14.36	[M–H]^−^	367.10	179, 135, 367, 161
4-coumaric acid	14.72	[M–H]^+^	309.09	147, 119, 91
Ferulic acid	15.05	[M–H]^+^	177.05	117, 89, 149, 145, 134
Epicatechin	15.20	[M–H]^−^	289.07	245, 205, 109, 125
Procyanidin Trimer	15.46	[M–H]^−^	865.20	125, 289, 407, 161
Coumaroylquinic acid	15.75	[M–H]^−^	377.09	173, 163
Quercetin-3-O-galactoside + cluster	15.93	[M–H]^+^	1443.30	163, 245, 257, 229
Quercetin-3-O-galactoside + cluster	16.55	[M–H]^+^	1443.30	163, 245, 257, 229
Quercetin-3-O-glucoside + cluster	16.94	[M–H]^+^	866.20	163, 245, 247, 257
Quercetin-3-O-glucoside + cluster	18.60	[M–H]^+^	866.20	163, 245, 247, 257
Quercetin-3-O-Rutinoside	19.82	[M–H]^−^	609.15	300, 302, 301, 564
Procyanidin B1 Dimer	21.14	[M–H]^−^	577.13	289, 125, 407, 245
Quercetin-3-O-galactoside	21.93	[M–H]^−^	463.09	300, 301
Quercetin-3-O-glucoside	22.59	[M–H]^+^	465.10	303, 229, 85, 153
Quercetin-3-O-xyloside	25.01	[M–H]^−^	433.08	300, 301
Quercetin-3-O-arabinoside	26.37	[M–H]^−^	433.08	300, 301
Quercetin-3-O-rhamnoside	27.75	[M–H]^−^	447.09	300, 301, 302, 447
Phloretin	28.04	[M–H]^+^	275.09	107, 169
Quercetin	29.19	[M–H]^−^	301.04	151, 179, 301
Quercitrin	30.08	[M–H]^+^	449.09	303, 71, 85, 137, 153
Phloridzin	37.48	[M–H]^−^	435.13	273, 167

**Table 2 plants-12-03397-t002:** Identification of phenolic compounds in pear samples using LC-ESI-MS/MS.

Proposed Compounds	RT (min)	Ionization (ESI^+^/ESI^−^)	Molecular Ion	MS/MS
Arbutin	9.75	[M–H]^−^	317	109, 109, 123
Quinic acid	10.37	[M–H]^−^	191	191
3′Caffeoylquinic acid	11.97	[M–H]^+^	355	355, 191
Syringic acid	12.67	[M–H]^−^	359	197, 153, 331
Protocatechuic acid	14.94	[M–H]^−^	153	109, 153
Feroloylquinic acid	16.77	[M–H]^−^	658	191, 193
Esculin	17.98	[M–H]^+^	341	179, 123
Caffeic acid hexoside	19.97	[M–H]^−^	341	179
Cholorogenic acid	20.56	[M–H]^−^	355	163
Umbeliferone	20.97	[M–H]^+^	163	163, 135
Catechin	21.08	[M–H]^−^	289	245, 289
5′Caffeoyl quinic acid	21.39	[M–H]^+^	355	163
Caffeic acid	21.76	[M–H]^−^	367.1	179, 135, 367, 161
Procyanidin dimer	22.14	[M–H]^−^	577	289, 407, 125
Myricetin-3-galactoside	22.99	[M–H]^−^	480	287, 317
Epicatechin gallate	23.61	[M–H]^−^	469	289, 245, 135
4′Caffeoyl quinic acid	24.36	[M–H]^−^	353	191
Epicatechin	24.83	[M–H]^−^	289	289, 125, 245
Coumaroyl quinic acid	25.05	[M–H]^−^	337	191, 93, 173
Procyanidin derivative	25.58	[M–H]^−^	757	125, 289, 407
Hydroxyferulic acid	25.89	[M–H]^−^	327	165
3-Feruloylquinic acid	26.26	[M–H]^+^	367/163	163
Caffeyolquinic acid	27.87	[M–H]^+^	367	163
Procyanidin derivative	28.03	[M–H]^−^	741	289, 339, 177
Quercetin-3-O-rutinoside	28.45	[M–H]^−^	609	300, 301, 609
Quercetin-3-O-galactoside	29.40	[M–H]^−^	463	300, 301, 463
Quercetin-3-O-glucoside	29.68	[M–H]^−^	463	302, 301, 463
Procyanidin derivative	30.53	[M–H]^−^	483	289, 245, 341
Isorhamnetin-3-O-rutinoside	30.67	[M–H]^+^	625	317
Isorhamnetin	30.86	[M–H]^−^	623	315
Coumaroyl hexoside	31.24	[M–H]^−^	351	163, 119
3,4-Dicaffeyolquinic acid	31.92	[M–H]^−^	515	191, 179, 70, 353
Isorhamnetin-3-O-glucoside	32.18	[M–H]^−^	477	314, 315, 477
Coumaroyl hexoside	32.62	[M–H]^−^	351	163, 119
Luteolin-4′-O-glucoside	34.00	[M–H]^−^	533	285, 284
Isorhamnetin acetyl hexosided	34.21	[M–H]^−^	519	314, 315, 299, 300, 316
Caffeyoquinic acid derivative	37.89	[M–H]^+^	531	163
Kampferol	38.29	[M–H]^−^	287.12	287
Caffeic acid methyl ester	53.34	[M–H]^−^	193	193
3,4,5-tricaffeoylquinic acid	58.02	[M–H]^+^	677	677, 515

**Table 3 plants-12-03397-t003:** Samples provided using SUMOL+COMPAL and their processing methods.

Fruit	Sample	Method of Production	Abbreviation
**Apple**	Apple	-	A
	Puree	With Colloidal milling using Urschel^®^	PWU
	Puree	Without Colloidal milling using Urschel^®^	P
	Pomace	Without Colloidal milling using Urschel^®^ Differential speed: 10 rpm at 70 °C	PO
	Pomace	With Colloidal milling using Urschel^®^ Differential speed: 10 rpm at 70 °C	POWU
	Turbid juice	Differential speed: 10 rpm at 70 °C	TJ10
	Turbid juice	Differential speed: 35 rpm at 70 °C	TJ35
	Clarified juice	Tangential filtration at 40 °C	CJ40
	Clarified juice	Tangential filtration at 60 °C	CJ60
	Pear	-	PE
**Pear**	Puree	With Colloidal milling using Urschel^®^	PPWU
	Puree	Without Colloidal milling using Urschel^®^	PP
	Pomace	Without Colloidal milling using Urschel^®^ Differential speed: 10 rpm at 70 °C	POP
	Turbid juice	Differential speed: 10 rpm at 70 °C	TJP10
	Turbid juice	Differential speed: 35 rpm at 70 °C	TJP35
	Clarified juice	Tangential filtration at 40 °C	CJP40
	Clarified juice	Tangential filtration at 60 °C	CJP60

## Data Availability

The data presented in this study are available on request from the corresponding author. The data are not publicly available due to privacy statements.

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
