# Peer review of "Characterization of the Effect of a Novel Production Technique for ‘Not from Concentrate’ Pear and Apple Juices on the Composition of Phenolic Compounds"

_plants, 2023, doi:10.3390/plants12193397_

Round 1

Reviewer 1 Report

Although the article is interesting and the research is well designed and conducted, it needs major revision to make it acceptable for publication in the journal. my comments are below:

1- Abstract is long, can be shorten by focusing on the most important findings. It contains detailed description to the methods used. I suggest removing the details and keeping the outline breifly.

2- The introduction is too long, I suggest reduction to the half. Concentrate on the date relevant to the abjectives and findings.

3- The theme you are working on doesn't need  theoretical data on phenolic classification, so I suggest removing of Figure.1

4- Line 129-138. You should state your objectives clearly and breifly, do not use any general data or information.

5- The discussion is just descriptive and superficial. I missed the comparison of your results (particularly the quantitative data) with those found by other researchers, how do they comapre?

6- Line216. You used TPC to express the total phenols content by Folin-Ciocalteu and HPLC methods, but you didn't differentiate them in the text, so use FC-TPC and HPLC-TPC whereever they appearedin the text.

7- Line 263-274. This paragraph is suitable for introduction, it is not a discussion of your results. Please delete it.

8- Line 318-324. It is not needed at all to explain why FC-TPC is different from HPLC-TPC. It is well known fro the food chemists and analysts, it doen't add any value for the discussion of your results. Remove it here and elsewhere.

9- Line 288-290, and line 341-343. It is not clear what do you want to say. Do you mean " the products is lower by 1-4% and 2-3% as compared to that in puree for example??

10- Line 345-364. It is repeated once again as for apple matrices,

11- line 350. Let this part to be quantitative by mentioning the concentration found by you and that found by the literature, and by what % it is higher or lowe.

12- There is no uniformity in the use of cited literature, sometimes you use author name et al..... and in other parts you used the DOI code!!!!

13- It is not clear how the major phenolic acids, and phloridzin relate to fingerprint, it should be discussed and stated correctly.

14- I suggest removing figure 8, since the data in table 5 are enough, the same suggestion for Figure 9. The two figures do not add any more interest and scientific value to the article.

15_ I missed the necessary references for the HPLC separation and HPLC-MS/MS identification methods, since thea are not your own methods.

my seriou problem is with the English phrase, expression and style. The authors are asked to look at any published aricle in the journal in order to use the correct phrases and style. I tried to correct some of the incorrect words or phrases:

Line 19: will focuse should write fucosed.

Line 141: Remove "The total phenolic content (TPC), let the sentence stats with Figure 2....

Line 153: use the next fruit processing....

Line 154: Remove " Namely"

Line 181: it should be written as. while with 10 rpm there was no any significant....

Line 208 and elsewhere: Remove " Looking at the chromatogram and the identification data, start the sentence with " The major phenolic....."

Line 213: by using , remove a

Line 216: remove quantified afzter TPC

line 219: insteade of composition use content, and insteade of However use although, correct leads to resulted ina a lower TPC.

Line 240: Remove the first part of the sentence " Looking at the characterization data"

line 242: Remove Overall.

Line 243: remove families.

Line 245: The sum of all comounds (not each comounds).

Line 251. obtained with (not for).

Line 312: has already reported ( give reference here).

Line 458: correct acidulated to acidified.

Line 474. add for after 20000 g

Line 523: change to hlp to for.

Line 555: chanhe "obtain" to produce or give.

Line 559: change " growth" to increase.

Line 562: change " to understand" to check or highlight .

General comment: The verbs in the article should be in the part time, while the verbs with cited data from literature should be in the present perfect. The article should be lectured from the Rnglish structure, spelling, and phrase.

Author Response

Dear reviewer, 

I appreciate your time and suggestions in order to improve the scientific quality of the manuscript. The manuscript was changed accordingly as follows:

Although the article is interesting and the research is well designed and conducted, it needs major revision to make it acceptable for publication in the journal. my comments are below:

1- Abstract is long, can be shorten by focusing on the most important findings. It contains detailed description to the methods used. I suggest removing the details and keeping the outline breifly.

The abstract was conveniently changed according the reviewer´s suggestions

2- The introduction is too long, I suggest reduction to the half. Concentrate on the date relevant to the abjectives and findings.

The introduction was significantly reduced to concentrate the context and objectives

3- The theme you are working on doesn't need  theoretical data on phenolic classification, so I suggest removing of Figure.1

The Figure 1 and some additional information regarding theoretical data on phenolic classification and structure were removed.

4- Line 129-138. You should state your objectives clearly and breifly, do not use any general data or information.

The objectives appear now clearly described.

5- The discussion is just descriptive and superficial. I missed the comparison of your results (particularly the quantitative data) with those found by other researchers, how do they comapre?

The discussion was improved and some additional information was added

6- Line216. You used TPC to express the total phenols content by Folin-Ciocalteu and HPLC methods, but you didn't differentiate them in the text, so use FC-TPC and HPLC-TPC whereever they appearedin the text.

The terminology suggested was included in the manuscript

7- Line 263-274. This paragraph is suitable for introduction, it is not a discussion of your results. Please delete it.

The paragraph was removed according the reviewer suggestion

8- Line 318-324. It is not needed at all to explain why FC-TPC is different from HPLC-TPC. It is well known fro the food chemists and analysts, it doen't add any value for the discussion of your results. Remove it here and elsewhere.

The difference between FC-TPC and HPLC-TPC can be explained by different hypothesis. It is clear and well known from the food chemists and analysts that each technique has their own limitations. However, we consider relevant to include this information because in this specific case the difference is not given by the presence of interferences which can affect the spectrophotometric measurement. The main difference is given by the presence of phenolics, the non-extractable fraction of phenolics which has tentatively analysed within this study but unfortunately there was not possible the complete identification/characterization.

9- Line 288-290, and line 341-343. It is not clear what do you want to say. Do you mean " the products is lower by 1-4% and 2-3% as compared to that in puree for example??

The sentence was reformulated

10- Line 345-364. It is repeated once again as for apple matrices,

The discussion was reformulated

11- line 350. Let this part to be quantitative by mentioning the concentration found by you and that found by the literature, and by what % it is higher or lowe.

The quantitative data was included

12- There is no uniformity in the use of cited literature, sometimes you use author name et al..... and in other parts you used the DOI code!!!!

The references were reviewed and uniformed. Including the DOI code was a mistake.

13- It is not clear how the major phenolic acids, and phloridzin relate to fingerprint, it should be discussed and stated correctly.

The phenolic compounds profile can be used as a fingerprint for fruit analysis (http://dx.doi.org/10.1016/j.foodchem.2022.132612). However, in order to avoid any misunderstanding, this sentence was reformulated to delete the fingerprint-related data.

14- I suggest removing figure 8, since the data in table 5 are enough, the same suggestion for Figure 9. The two figures do not add any more interest and scientific value to the article.

The reader can have additional information in the Figures since include information about the chromatographic profile. The relative amount of each identified peak can be extracted from Figures and this information is missing in the Table.

15_ I missed the necessary references for the HPLC separation and HPLC-MS/MS identification methods, since thea are not your own methods.

Actually, there are methods properly optimized in our laboratory.

Comments on the Quality of English Language

my seriou problem is with the English phrase, expression and style. The authors are asked to look at any published aricle in the journal in order to use the correct phrases and style. I tried to correct some of the incorrect words or phrases:

Line 19: will focuse should write fucosed.

Focused

Line 141: Remove "The total phenolic content (TPC), let the sentence stats with Figure 2....

The sentence was altered

Additionally, all suggestions as well additional improvements were done to improve the scientific expression and English language as follows:

Line 153: use the next fruit processing....

Line 154: Remove " Namely"

Line 181: it should be written as. while with 10 rpm there was no any significant....

Line 208 and elsewhere: Remove " Looking at the chromatogram and the identification data, start the sentence with " The major phenolic....."

Line 213: by using , remove a

Line 216: remove quantified afzter TPC

line 219: insteade of composition use content, and insteade of However use although, correct leads to resulted ina a lower TPC.

Line 240: Remove the first part of the sentence " Looking at the characterization data"

line 242: Remove Overall.

Line 243: remove families.

Line 245: The sum of all comounds (not each comounds).

Line 251. obtained with (not for).

Line 312: has already reported ( give reference here).

Line 458: correct acidulated to acidified.

Line 474. add for after 20000 g

Line 523: change to hlp to for.

Line 555: chanhe "obtain" to produce or give.

Line 559: change " growth" to increase.

Line 562: change " to understand" to check or highlight .

General comment: The verbs in the article should be in the part time, while the verbs with cited data from literature should be in the present perfect. The article should be lectured from the Rnglish structure, spelling, and phrase.

Reviewer 2 Report

"Not from concentrate" juice is a new trend as the novel and healthy food, this manuscript investigate the phenolic profile based on the processing conditions. There are some suggestions for the quality improvement.

1. Line 15, typo "Ppolyphenols"

2. LIne 19 "will focus" to "is focused"

3. Line 101, please check the sentence.

4. Line 108, extra period observed.

5. Lines 115-117, Lines 127-128, please check the clarity of these sentences.

6. The unknown compounds can be not listed on the Table 1 & 2.

7. Line 434, What is "SUMOL+COMPAL", a company? A farm or a research institute?

Some sentences are a little bit difficult to follow. The author should check the lauguage expression and make the improvement.

Author Response

Dear reviewer, 
I appreciate your time and suggestions in order to improve the scientific quality of the manuscript. The manuscript was changed accordingly as follows:

"Not from concentrate" juice is a new trend as the novel and healthy food, this manuscript investigate the phenolic profile based on the processing conditions. There are some suggestions for the quality improvement.

  1. Line 15, typo "Ppolyphenols"
  2. LIne 19 "will focus" to "is focused"

Some changes were included in the abstract according to additional reviewers and this errors were properly corrected

  1. Line 101, please check the sentence.

The sentence was checked

  1. Line 108, extra period observed.

There was corrected

  1. Lines 115-117, Lines 127-128, please check the clarity of these sentences.

The phrases were reformulated

  1. The unknown compounds can be not listed on the Table 1 & 2.

The unknown compounds were included because some structural information is displayed and can serve as the basis for future work helping in the identification. Likewise, the unknown compounds are minor compounds and can be highlighted in the Figures.

  1. Line 434, What is "SUMOL+COMPAL", a company? A farm or a research institute?

Sumol+compal is a company. Some additional information was included

Reviewer 3 Report

The manuscript was built on a case-study, in which different samples were collected from a juice-producer, being then analysed using Folin-Ciocalteu method, as well as HPLC and LC-MS/MS .

Unfortunately the manuscript has numerous drawbacks, has an unkempt structure, is too wordy and in many cases English raises comprehension issues, hence the manuscript needs a major revision. Please consider the following issues for an improved version:

English needs improvement

L.15 – change “ Ppolyphenols” > polyphenols

L.19 – use Past Tense in the manuscript when referring to the accomplished research – this is applicable also for next instances (e.g.L.129 – 137);

L.19 – rephrase “procuded”

L.26 – wrong technical language; replace “ of DW” > DW, as well as all the next instances of this!

L.31 – wrong translation -  tangencial – revise

L.34 – 35 – incomplete phrase “ concentrated juices produced  from concentrated….”

L.55 – delete “indeed” – filler; avoid the annoying repetition of this word and replace it with synonyms or better delete it in next instances (L.55, 82, 96, 125, 174,

L.72 – 73 – delete “Phenolic compounds 71 result from the secondary metabolism of plants” – filler, basic knowledge

L.80-89 – this paragraph contains mostly basic knowledge; delete all unnecessary text, as well as Figure 1 (fillers)

L.98, 106, 177, 183, etc – avoid the annoying repetition of  “Likewise”

L.98 – Latin names should be in italics

L.101 – rephrase “rom”

L.104 – delete “namely” - filler

L.142 – rephrase (repetition)

L.142 – 145 – avoid duplication of acronyms – you have them explained also in figure 2

L.146 – improper technical language “two forces” – rephrase

L.154 – delete “namely” - filler

#Figure 2 – pay attention to the units on Y-axis; remove all stars and connecting lines

L.69– 173– avoid duplication of acronyms – you have them explained also in figure 3

L.65, 180, 181, 189, 190, 226 – inaccurate technical language – rephrase all instances of “velocity”,

#Figure 3 – pay attention to the units on Y-axis; remove all stars and connecting lines

#Figure 5–  wrong data – see the corresponding observation – this has to be deleted, as well as the related discussion

L.238 – table 2 contains “a bunch” of compounds, lot of then not phenolic compounds;  remove all unnecessary substances, from this table and update the peak indexes from the related figure

L.242 – 243 – delete “Overall, the phenolic compounds found in pear belong to the phenolic acids and flavonols families.” – filler

#Figure 7–  wrong data – see the corresponding observation – this has to be deleted, as well as the related discussion

#3 – move all common knowledge text in introduction or delete it and keep only discussion related to your data here (e.g. L.264-281); delete all filler repetitions (e.g. L.279-281)

L.429 – rephrase “Chloridiric acid”

L.430 – what about sodium carbonate?

#4.2 – since the products analysed in this context originates from an industrial producer, you have to give all the relevant technological parameters, as well as a proper description of the processing stages; replace the childlike figure 8 with a decent scheme (ask an engineer for it…)

L.464 – 467 – remove all un-necessary text and provide a descriptive one

#4.5 – fruit samples have to be sorted, pre-processed (e.g. peeled, cut in pieces… weighed…) – missing descriptors for these stages

L.47 4 – wrong unit min-1; add the centrifuge type and producer

L.47 5 – add the rotavapor type and producer

L.490 – replace acronyms DE/EA with their meaning

L.492- 493 – what is the meaning of “Since there was no possible to identify the bounded phenolics.”

#4.7 – this is a spectrophotometric method; add the instrument type & producer, the wavelength used for measurement, as well as the calibration range

#4.8 why do the mobile phases differ for different matrices?

L.519-523 – move this section  under #4.1 and add the provider

L.523 – 526 – improper chromatographic data processing; since the authors reported they use reference standards for chromatographic analysis, they have to properly calibrate the system in order to obtain adequate results for each compound

L.543 – define the acronym SEM

#5 Describe in 1-2 phrases what the contribution of this research is.

 Extensive editing of English language required - in many cases English raises comprehension issues.

Author Response

Dear reviewer, 
I appreciate your time and suggestions in order to improve the scientific quality of the manuscript. The manuscript was changed accordingly as follows:

The manuscript was built on a case-study, in which different samples were collected from a juice-producer, being then analysed using Folin-Ciocalteu method, as well as HPLC and LC-MS/MS .

Unfortunately the manuscript has numerous drawbacks, has an unkempt structure, is too wordy and in many cases English raises comprehension issues, hence the manuscript needs a major revision. Please consider the following issues for an improved version:

English needs improvement

L.15 – change “ Ppolyphenols” > polyphenols

The typographic error was corrected

L.19 – use Past Tense in the manuscript when referring to the accomplished research – this is applicable also for next instances (e.g.L.129 – 137);

The changes were included

L.19 – rephrase “procuded”

The typographic error was corrected

L.26 – wrong technical language; replace “ of DW” > DW, as well as all the next instances of this!

The changes were included

L.31 – wrong translation -  tangencial – revise

The typographic error was corrected

L.34 – 35 – incomplete phrase “ concentrated juices produced  from concentrated….”

The sentence was reformulated

L.55 – delete “indeed” – filler; avoid the annoying repetition of this word and replace it with synonyms or better delete it in next instances (L.55, 82, 96, 125, 174,

The sentence was reformulated

L.72 – 73 – delete “Phenolic compounds 71 result from the secondary metabolism of plants” – filler, basic knowledge

The information was removed

L.80-89 – this paragraph contains mostly basic knowledge; delete all unnecessary text, as well as Figure 1 (fillers)

The information was removed

L.98, 106, 177, 183, etc – avoid the annoying repetition of  “Likewise”

The paragraph was reformulated

L.98 – Latin names should be in italics

Latin names were written in italics

L.101 – rephrase “rom”

The typographic error was corrected

L.104 – delete “namely” – filler

The word was deleted

L.142 – rephrase (repetition)

The sentence was reformulated

L.142 – 145 – avoid duplication of acronyms – you have them explained also in figure 2

To our understanding, including the acronyms in each figure, makes easy the interpretation of the results.

L.146 – improper technical language “two forces” – rephrase

The term was replaced by centrifugal force/speed (both are technically correct)

L.154 – delete “namely” – filler

The word was removed

#Figure 2 – pay attention to the units on Y-axis; remove all stars and connecting lines

The stars and connecting lines showed the statistical analysis so they can be present in the Figure in order to validate the significance of the values

L.69– 173– avoid duplication of acronyms – you have them explained also in figure 3

To our understanding, including the acronyms in each figure, makes easy the interpretation of the results.

L.65, 180, 181, 189, 190, 226 – inaccurate technical language – rephrase all instances of “velocity”,

The term velocity was replaced by speed

#Figure 3 – pay attention to the units on Y-axis; remove all stars and connecting lines

The stars and connecting lines showed the statistical analysis so they can be present in the Figure in order to validate the significance of the values

#Figure 5–  wrong data – see the corresponding observation – this has to be deleted, as well as the related discussion

After verifying the data included in Figure 5, we cannot realise which wrong data are you referring to? Can you please give us some concrete information in order to verify if there is any mistake?

L.238 – table 2 contains “a bunch” of compounds, lot of then not phenolic compounds;  remove all unnecessary substances, from this table and update the peak indexes from the related figure

Some organic acids were also identified. There are not phenolic compounds but they are identified after phenolic compounds extraction. A sentence explaining this was added to the manuscript.

L.242 – 243 – delete “Overall, the phenolic compounds found in pear belong to the phenolic acids and flavonols families.” – filler

The sentence was removed

#Figure 7–  wrong data – see the corresponding observation – this has to be deleted, as well as the related discussion

After verifying the data included in Figure 7, we cannot realise which wrong data are you referring to? Can you please give us some concrete information in order to verify if there is any mistake?

#3 – move all common knowledge text in introduction or delete it and keep only discussion related to your data here (e.g. L.264-281); delete all filler repetitions (e.g. L.279-281)

The discussion was improved.

L.429 – rephrase “Chloridiric acid”

The typographic error was corrected

L.430 – what about sodium carbonate?

The company where was purchased was added

#4.2 – since the products analysed in this context originates from an industrial producer, you have to give all the relevant technological parameters, as well as a proper description of the processing stages; replace the childlike figure 8 with a decent scheme (ask an engineer for it…)

The figure included in the manuscript was not and industrial flow diagram. The figure was now replaced

L.464 – 467 – remove all un-necessary text and provide a descriptive one

Some descriptive text was added.

#4.5 – fruit samples have to be sorted, pre-processed (e.g. peeled, cut in pieces… weighed…) – missing descriptors for these stages

The fruit were weighted, washed and directly crushed as included in the diagram.

L.47 4 – wrong unit min-1; add the centrifuge type and producer

This information was added

L.47 5 – add the rotavapor type and produce

This information was added

L.490 – replace acronyms DE/EA with their meaning

The acronyms were replaced

L.492- 493 – what is the meaning of “Since there was no possible to identify the bounded phenolics.”

The phenolic compounds can appear in nature bounded mainly to fibre resulting in non-extractable phenolics.

#4.7 – this is a spectrophotometric method; add the instrument type & producer, the wavelength used for measurement, as well as the calibration range

The instrument type was already included in Line 457, as well as the wavelength used. The calibration range was now included.

#4.8 why do the mobile phases differ for different matrices?

The solvents and gradient conditions were optimized for each matrix in order to obtain the best resolution conditions.

L.519-523 – move this section  under #4.1 and add the provider

The information was reorganized following the suggestions.

L.523 – 526 – improper chromatographic data processing; since the authors reported they use reference standards for chromatographic analysis, they have to properly calibrate the system in order to obtain adequate results for each compound.

The use of standards is quite limited in the identification of phenolic compounds. Indeed, standards or a standard representing a specific family of phenolic compounds are usually used for quantification purposes more than for identification. There are more than 8000 structures already described and there are not commercially available standards of each compound. Under this context, and considering that the characterization was performed in an orbitrap, the identification was performed by using the exact mass and the fragmentation pattern and the compounds were identified in equivalents of the major compound present in apple as previously published.

L.543 – define the acronym SEM

SEM was replaced by the extended form.

#5 Describe in 1-2 phrases what the contribution of this research is.

Conclusions were improved by adding 1-2 phrases indicating the contribution of study to food science and food industry.

Reviewer 4 Report

In the reviewed article, the authors present the work of characterising the effect of a novel production technique for NFC juices on the phenolic compounds in apples and pears. The proposed study confirmed that NFC juices are good options for obtaining juices with more polyphenols.

In my opinion, the article cannot be accepted for publication.

The manuscript is generally well written (however, some sentences couldn’t understand – p.4 lines 142-144, missing words – p4 line 142, too much space, double dots, missing space etc. ), but the presented study does not improve the knowledge about the proposed topic. In the publication, we could find only information about confirming the presence of polyphenols in juice. Still, this information is known and described in the literature (Candrawinata, Vincent & Blades, Barbara & Golding, John & Stathopoulos, Costas & Roach, Paul. (2012). Effect of clarification on the polyphenolic compound content and antioxidant activity of commercial apple juices. International Food Research Journal. 19. 1055-1061; Brunda G, Kavyashree U, Shetty SS, Sharma K. Comparative study of not from concentrate and reconstituted from the concentrate of pomegranate juices on nutritional and sensory profile. Food Science and Technology International. 2022;28(1):93-104; JarosÅ‚aw Markowski, Alain Baron, Jean-Michel Le Quéré, Witold PÅ‚ocharski, Composition of clear and cloudy juices from French and Polish apples in relation to processing technology, LWT - Food Science and Technology, 62, 2015,  813-820)

Main concerns:

1)     The analytical grade reagents for LC-MS/MS analysis are unacceptable.

2)     I couldn’t find information on how the compounds were identified by LC-MS/MS analysis. Did you use some standards? Did you identify by Product ion analysis?

Author Response

Dear reviewer, 
I appreciate your time and suggestions in order to improve the scientific quality of the manuscript. The manuscript was changed accordingly as follows:

In the reviewed article, the authors present the work of characterising the effect of a novel production technique for NFC juices on the phenolic compounds in apples and pears. The proposed study confirmed that NFC juices are good options for obtaining juices with more polyphenols.

In my opinion, the article cannot be accepted for publication.

The manuscript is generally well written (however, some sentences couldn’t understand – p.4 lines 142-144, missing words – p4 line 142, too much space, double dots, missing space etc. ), but the presented study does not improve the knowledge about the proposed topic. In the publication, we could find only information about confirming the presence of polyphenols in juice. Still, this information is known and described in the literature (Candrawinata, Vincent & Blades, Barbara & Golding, John & Stathopoulos, Costas & Roach, Paul. (2012). Effect of clarification on the polyphenolic compound content and antioxidant activity of commercial apple juices. International Food Research Journal. 19. 1055-1061; Brunda G, Kavyashree U, Shetty SS, Sharma K. Comparative study of not from concentrate and reconstituted from the concentrate of pomegranate juices on nutritional and sensory profile. Food Science and Technology International. 2022;28(1):93-104; JarosÅ‚aw Markowski, Alain Baron, Jean-Michel Le Quéré, Witold PÅ‚ocharski, Composition of clear and cloudy juices from French and Polish apples in relation to processing technology, LWT - Food Science and Technology, 62, 2015,  813-820)

Main concerns:

  • The analytical grade reagents for LC-MS/MS analysis are unacceptable.

The solvents used are of LC-MS/MS grade, this was a mistake. A sentence including the specificity in solvents quality depending on the purposes was added.

  • I couldn’t find information on how the compounds were identified by LC-MS/MS analysis. Did you use some standards? Did you identify by Product ion analysis?

The use of standards is quite limited in the identification of phenolic compounds. Indeed, standards or a standard representing a specific family of phenolic compounds are usually used for quantification purposes more than for identification. There are more than 8000 structures already described and there are not commercially available standards of each compound. Under this context, and considering that the characterization was performed in an orbitrap, the identification was performed by using the exact mass and the fragmentation pattern. The available standards were additionally used as described in section 4.8. Some additional information was included in order to clarify any misunderstanding.

Round 2

Reviewer 1 Report

No more comments

Moderate improvement is still needed on the MS, e.g. as showing to be corrected to as shown etc.

Author Response

Thank you for your suggestions. The MS was revised and some spelling and/or gramatical errors were corrected

Reviewer 2 Report

I recommend accepting it under the current version.

Author Response

Thank you very much for accepting the manuscript in the current form

Reviewer 4 Report

I have no more comments

Author Response

Thank you very much for your suggestions. Some additional changes were performed in order to properly improve the scientific quality of the manuscript. Additional standards were used for quantification purposes, some additional information was included to properly understand the identification tasks, some figures were removed or changed, the phenolic compounds were grouped by families in the cromatographic analysis and the gramar was revised in the whole manuscript